# Setting of Natural Fracture Splitting Surface on Connecting Rod and Its Formation Mechanism

**Fengjun Zhang * and Yi Zhang**

College of Mechanical and Electrical Engineering, Nanjing University of Aeronautics and Astronautics,
Nanjing 210016, China; jszhangyi@yahoo.com

**\*** Correspondence: jszhfj@nuaa.edu.cn; Tel.: +86-1876-168-5367

**Abstract:** To break through the limitation of fracture splitting process on material selection and solve problems during fracture splitting such as parts tearing, failing to split, dropping dregs, fracture surface deformation and so on, a new technique of setting natural splitting surface in casting blank is proposed, aiming to achieve brittle fracture along pre-set surface during fracture splitting process. In this research, casting blanks are produced with metal molds. A layer of AZ31 foil is set in advance before casting, the layer interacts with liquid LD10 aluminum alloy, forming a brittle interface layer across the whole casting, then a fracture splitting hole is machined in the middle of the casting blank and cracking grooves are machined on the inner hole near the interface to achieve fracture splitting. Experiment revealed that the initial crack on the specimen starts from the root of the cracking groove, and the crack basically expands along the pre-set fracture splitting surface. The fracture surface is characterized by flaky brittle fracture. There is residual magnesium and pellumina, which have characteristics of melt with low-melting point, and micro-porosity in the fracture. Further analysis suggests that the formation mechanism of a natural fracture splitting surface can be described as follows: the magnesium foil with strong oxidation in high-temperature alloy liquid interacts with the pellumina at the front of liquid flow, which forms a interface. Meanwhile a layer of melt with a low-melting point forming as a result of interface reaction is pushed to the edge of the grain boundary, and surface liquid film shrinks to be micro-porosity. With such a combined effect it finally forms the brittle surface, which provides the condition for conducting subsequent fracture splitting process.

**Keywords:** fracture splitting process; setting of fracture splitting surface; splitting parts; brittle fracture

---

## 1. Introduction

Fracture splitting technique is the most advanced processing technique with high precision for manufacturing splitting parts, which is widely applied in fields such as engine connecting rods and crankcase covers. According to the fracture theory, specific cracking grooves are machined in advance on blanks to form a source of stress concentration, expansion fracture load is applied to make crack expand directionally, which achieves fracture without cuttings, and finally completes the fracture splitting process of connecting rod body and its cover [1,2]. There is no need to machine fracture surface further on a separated part, but to use the natural three-dimensional concave-convex structure on the fracture surface of connecting rod to achieve precise assembly between connecting rod body and its cover [3,4]. Compared to the traditional saw cutting method, the fracture splitting technique cancels mechanical sawing and the grinding process of joint surface on connecting rod body or its cover, and utilizes natural three-dimensional concave-convex surface forming through fracture to obtain accurate position of the joint surface on connecting rod in three directions. After assembly, the connecting rod body and its cover can contact each other precisely and lock each other, which greatly improves bearing capacity, shear capacity, and rigidity of the connecting rod. Meanwhile, high meshing performance

and assembly accuracy during re-assembly of the connecting rod can be still guaranteed. The fracture splitting technique simplifies structure design of bolt holes and eliminates fine machining process of the bolt holes, and, additionally, it saves investment in machine tools, reduces consumption of various kinds of cutting tools and energy dissipation, and decreases manufacturing cost of products [5,6]. This technology has been widely used in automobile engine connecting rod processing [7,8].

The fracture splitting technique is strict in material selection. On the premise of guaranteeing comprehensive mechanical properties of the connecting rod, toughness index is limited. Material is supposed to possess characteristic of brittle fracture at room temperature, as well as sufficient tensile strength on fracture surface, which severely limits the optional fracture splitting materials. At present, the optional fracture splitting materials in engineering field are limited, which mainly includes high carbon steel, Powder metallurgy materials such as Cu, Cr, Zr based powder, malleable cast iron, etc. [9,10], while, materials with high toughness such as ordinary steel, aluminum alloy, titanium alloy and so on generally cannot be applied in fracture splitting technique. At the present stage, mature technique of preparing fracture splitting materials is complicated, and demand of material composition control is very strict, which directly restricts the application and promotion of the fracture splitting technique [11]. In addition, various problems exist in fracture splitting process of common materials, such as fracture splitting deformation, fracture surface displacement, fracture surface excursion, bifurcation, one-side tearing, open seam, and so on [12]. To overcome the mentioned problems, domestic and oversea researches mainly focus on explorations including micro-alloy processing to improve material properties, development of new fracture splitting equipment, as well as optimization of fracture splitting parameters and so on. However, these are unable to break through the limitation of fracture splitting technique on material selection.

In order to solve the problems existing in fracture splitting technique, a new technique of setting natural fracture splitting surface in blanks is proposed, aiming to achieve brittle fracture splitting of parts along a pre-set surface in the fracture splitting process and guarantee fine meshing performance and sufficient strength of the two fracture surfaces, and provide location references for subsequent re-assembly of the splitting parts. Setting natural fracture splitting surface in blanks of connecting rods is a new technique with great innovation, which will accelerate the application and promotion of the fracture splitting technique. It provides practical technique for breaking through the limitation of fracture splitting technique on material selection, which is of great academic value and has large economic benefits.

## 2. Process Conditions and Experiment

### 2.1. Experimental Equipment

To make the brittle surface on a connecting rod pass throughout the entire casting section without being away from the pre-set location, a high demand of mold is required. A combined type metal mold is adopted in the experiment. A gating system has a single gate and double channels, namely a main channel and two sub-channels with symmetrical distribution, which divides metal liquid into two parts, flowing relatively into cavities, reaching each side of the foil almost at the same time, therefore filling two cavities smoothly. Ejection rods are set at the bottom of cavities to obtain smooth ejection of the casting, which is shown in Figure 1. Mutually matched convex-concave structure is set in mold, which is perpendicular to parting surface, so as to make it convenient for the clamping of AZ31 foil.

The technical way of setting a natural fracture splitting surface is that a layer of AZ31 foil is pre-set in the fracture splitting area. During casting, alloy liquid smoothly reaches each side of the foil almost at the same time, LD10 alloy liquid and the foil interact with each other and a brittle surface forms throughout the entire casting section where the AZ31 foil is set before. The initial crack appears as a result of stress concentration when fracture splitting loads are applied to the connecting rod blank with brittle interface. The crack extends along the brittle surface until a complete brittle fracture occurs.

Then, the interlocking structure on the fracture surface, which can mesh completely, is utilized to realize reposition and precise assembly between connecting rod body and its cover.

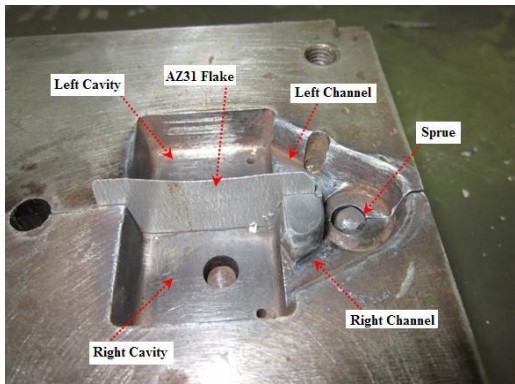

**Figure 1.** Mold device.

### 2.2. Natural Fracture Splitting Process Experiment

LD10 alloy with melting point of 660 °C is chosen as the material of the connecting rod. The chemical composition of LD10 is listed in Table 1. AZ31 foil with thickness of 0.12 mm and melting point of 649 °C is chosen as the magnesium alloy foil. The chemical composition of AZ31 (mass fraction %) is listed in Table 2. Oxidation reaction between molten LD10 alloy and AZ31 foil occurs easily under high temperature, producing Al-Mg crystalline oxide. Such oxide is left inside the casting after cooling, which plays an important role in forming brittle surface and lays a foundation for subsequently producing brittle fracture on connecting rod.

**Table 1.** Chemical composition of LD10 (mass fraction %).

| Mg | Cu | Si | Mn | Fe | Ti | Zn | Al |
|---------|---------|---------|-------|-------|-------|-------|--------|
| 0.4–0.8 | 3.9–4.8 | 0.6–1.2 | 0.4–1 | ≤0.7 | ≤0.15 | ≤0.3 | Remain |

**Table 2.** Chemical composition of AZ31 (mass fraction %).

| Al | Zn | Mn | Cu | Fe | Be | Si | Mg |
|---------|---------|-----|------|------|------|------|--------|
| 3.0–3.1 | 0.9–1.0 | 0.2 | 0.05 | 0.05 | 0.02 | 0.15 | Remain |

As shown in Figure 2, the experiment includes: (1) clamping the AZ31 foil, assembling and preheating the mold; (2) casting LD10 alloy liquid; (3) taking out the casting blank after cooling and then clearing the casting; (4) machining the center hole and cracking grooves on the inner hole near the interface; (5) implementing fracture splitting process for the specimen; and (6) observing fracture characteristics and testing meshing performance. A rectangular specimen with a size of 40 mm × 30 mm × 20 mm is obtained by wire-electrode cutting on the casting. A round hole with a diameter of φ14 mm is machined across the interface on specimen, V-shaped cracking grooves with a width of 1.5 mm and depth of 1 mm are machined on the inner hole near the interface. We then applied a bursting load through the cracking cone rod in the round hole to make it fracture along the brittle interface.

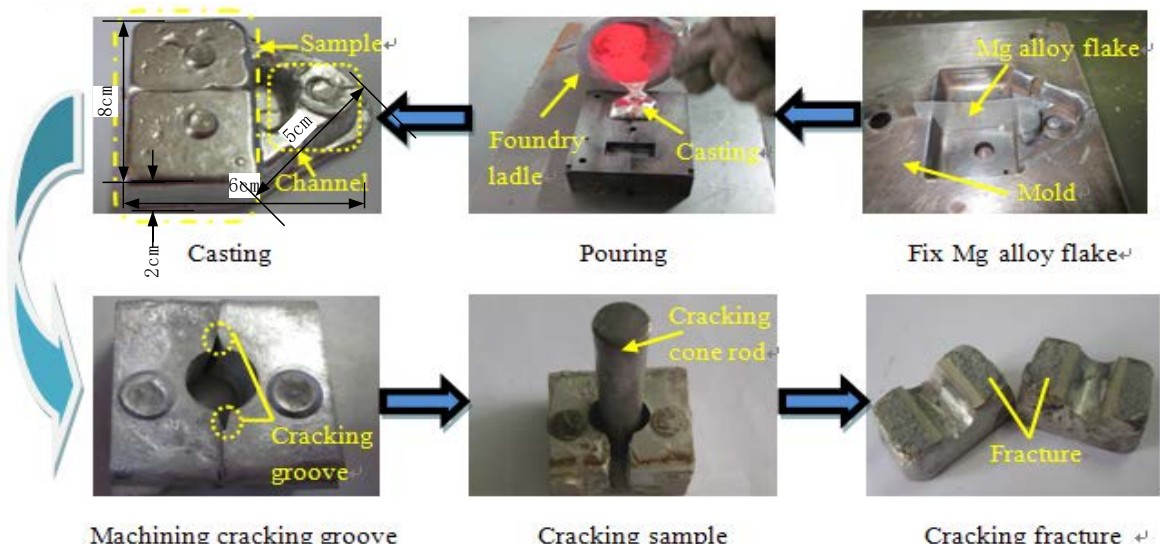

**Figure 2.** Flow diagram of fracture splitting process.

### 2.3. Influence of Magnesium Alloy Foil Thickness on the Quality of the Interface

To study the influence of the AZ31 foil thickness on the brittle interface, three foils with different thicknesses were chosen in experiment. Several castings with AZ31 foils of each thickness were prepared. The fusion situation of the foils and interface bonding quality are listed in Table 3. In Figure 3a it shows the surface fusion situation, while the thickness of magnesium foil is 0.3 mm. Since the foil had a relatively large thickness, it needed much more melting energy, coupled with the two aluminum liquid not fusing completely in the overlapped place. In Figure 3b two strands of aluminum alloy are not fused at the edges while Figure 3c shows the surface fusion situation with magnesium foil thickness of 0.12 mm, which has a clear bonding surface, and good bonding quality resulting from complete fusion between magnesium foil and aluminum base.

**Table 3.** Fracture splitting surface quality with AZ31 foils of different thicknesses.

| The AZ31 Foil Thickness (mm) | 0.12 | 0.24 | 0.30 |
|---|---|---|---|
| Casting temperature (°C) | 720 | 720 | 720 |
| Foil fusion state | Complete fusion | Failed fusion at the edge | Failed fusion |
| Fracture splitting surface bonding quality | High bonding strength | Failed bonding locally | Obvious overlap trace on surface |
| Fracture quality | Flaky brittle fracture | Poor fracture quality | Serious dropping dregs |

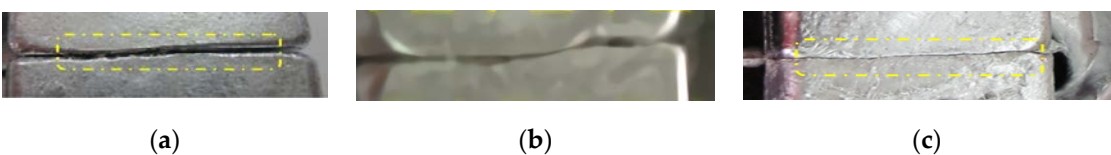

(**a**)  (**b**)  (**c**)

**Figure 3.** Casting bonding surfaces with AZ31 foil of different thicknesses (**a**) 0.3 mm AZ31 foil, (**b**) 0.24 mm AZ31 foil, (**c**) 0.12 mm AZ31 foil.

### 2.4. Influence of Casting Temperature on the Quality of the Interface

Casting temperature of LD10 alloy liquid generally ranges from 680 to 740 °C. Qualities of alloy organization and intermediate interface layers are different under different casting temperatures. According to the relevant composite theory, the entire process of bimetallic interface formation includes

three stages: physical contact formation, contact surface activation, and element diffusion. A high temperature can melt the metal and promote the full contact of the two alloys. At a high temperature, infiltration and diffusion occur, so that atoms between different elements migration occur to form a strong metallurgical bond [13]. To conduct the experiment, the mold is heated to 280 °C in advance and AZ31 foil with thickness of 0.12 mm is placed in the mold. Figure 4a shows the casting acquired under casting temperature of 770 °C, and it can be seen that the AZ31 foil in the intermediate position fused with the base metal completely and pre-set AZ31 foil combusted and oxidized completely. The interface forming under this temperature goes against the subsequent fracture splitting process. Figure 4b shows the casting acquired under a casting temperature of 670 °C, and it can be seen that fusion situation of the pre-set foil and base metal was relatively poor and the bonding surface did not fuse, which led to insufficient bonding strength of the casting and therefore failed to meet the requirements on mechanical performance of parts. Different situations of bonding surfaces forming in the castings under different casting temperatures are listed in Table 4.

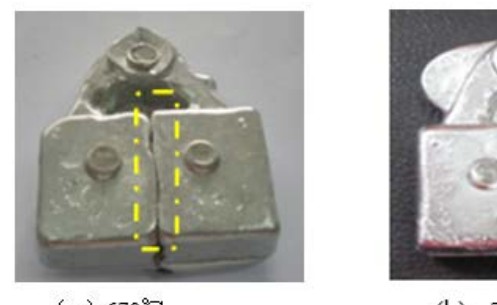
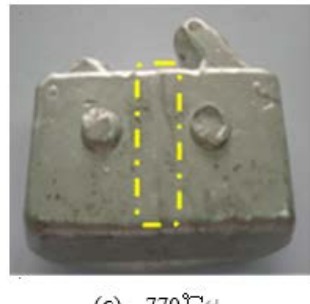

**Figure 4.** Casting bonding surfaces with different casting temperatures (**a**) 770 °C, (**b**) 670 °C, and (**c**) 720 °C.

**Table 4.** Fracture splitting surface quality with different casting temperatures.

| Casting Temperature (°C) | 670 | 720 | 770 |
|---|---|---|---|
| Casting situation | Without combustion of the AZ31 foil | Without combustion of the AZ31 foil | Combustion of the AZ31 foil |
| Fusion situation | Relatively large fusion trace | Relatively small fusion trace | Complete fusion |
| Temperature suitability | Relatively low | Suitable | Relatively high |

## 3. Fracture Surface Characteristics and Formation Mechanism of Fracture Splitting Surface

### 3.1. Fracture Surface Characteristics

The experiment shows that casting, which had pre-set AZ31 foil with thickness of 0.12 mm and was acquired under casting temperature of 720 °C, possesses good quality. Additionally, the brittle surface forming in casting is suitable for subsequent fracture splitting process. Castings prepared based on above conditions were added expansion load to conduct fracture splitting and specimens cracked along the bonding surface under expansion load. Characteristics of macroscopic fracture surface is characterized by a flat section and small deformation in plastic area, as shown in Figure 5b. With the same conditions, fracture splitting of base metal casting without AZ31 foil was implemented, as shown in Figure 5a. Compared with the former, there are obvious differences between their fracture surfaces, which mainly displays in: (1) the initial position of fracture splitting—the specimen with AZ31 foil produces initial crack from the root of cracking notch and the crack expands along the pre-set surface, forming a flat section without plastic deformation, while the initial position of crack on base metal is random and the section has large plasticity; (2) the fracture surface color—the fracture surface with AZ31 foil has gloom color while the fracture surface on the base metal is bright; (3) fracture surface deformation—the specimen with AZ31 foil needs small expansion fracture load, which is easy to realize fracture splitting separation, has no problems such as failing to split, dropping dregs, fracture

surface deformation and so on, and achieves low-energy brittle fracture, while the fracture surface on the base metal after fracture splitting is uneven and has large deformation, as well as requiring a large expansion fracture load.

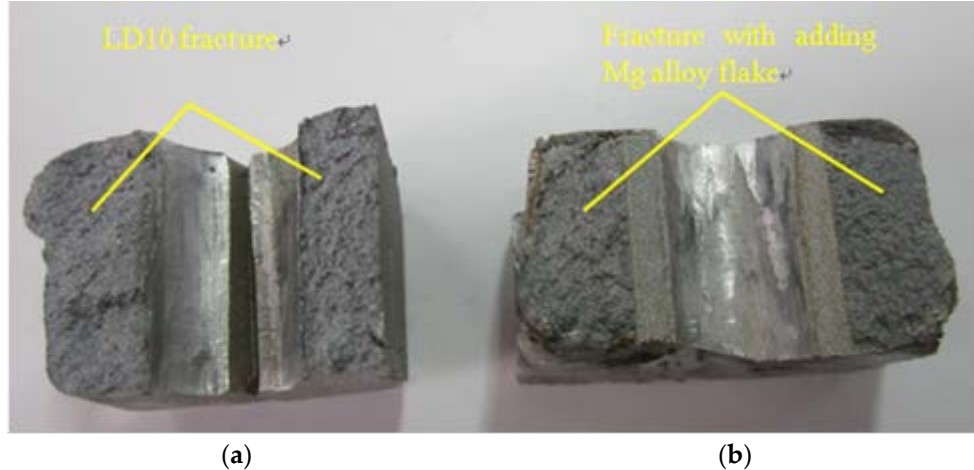

(**a**)                    (**b**)

**Figure 5.** Macro fracture after casting cracking. (**a**) LD10 base material macro fracture, (**b**) foil added casting macro fracture.

Scanning electron microscope is employed to observe fracture surface characteristics. Figure 6 shows the characteristics of a casting with pre-set natural fracture splitting surface. Fracture surface morphology is characterized by a flaky brittle fracture, and, locally, small plastic deformation at the edge of flaky fracture surface area exist, which obviously shows the characteristics of brittle fracture. As a contrast, microscopic morphology of fracture surface on base metal of the casting after fracture splitting is shown in Figure 7. From it we can see that the fracture surface has obvious plastic deformation and there are many ductile tearing traces, which belongs to typical ductile fracture.

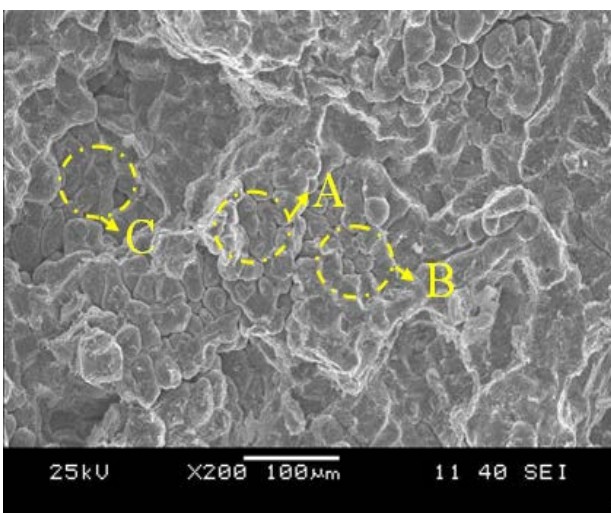

**Figure 6.** Macroscopic fracture surface of casting with AZ31 foil.

Compared with the above two fracture surfaces (Figure 5a,b, Figure 6, and Figure 7), it is found that the cracking along the natural fracture splitting surface are characterized by remarkable characteristics as follows: (1) brittle fracture; (2) intergranular fracture; and (3) flaky fracture surface. The technique of setting such fracture splitting surface meets the requirements on fracture splitting of connecting rods, which reduces plasticity in fracture splitting area on the connecting rod, and has a flat fracture surface without phenomena such as crack bifurcation and excursion.

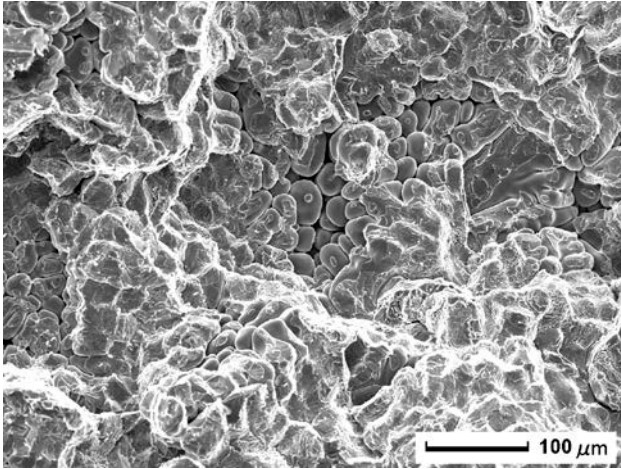

**Figure 7.** Macroscopic fracture surface of LD10 base metal.

*3.2. Analysis of Formation Mechanism of Fracture Splitting Surface*

According to inference based on fracture surface characteristics, the formation mechanisms of oxidation film fracture splitting surface, liquid film fracture splitting surface, and micro-porosity fracture splitting surface are analyzed in the following.

3.2.1. Formation of Oxidation Film Fracture Splitting Surface

When conducting casting, two relatively flowing LD10 liquid meet and overlap on each side of the AZ31 foil. Under high temperature, the foil is severely oxidated, which produces Al-Mg oxide. At the same time, pellumina at front of alloy liquid flow is pushed to the place near the foil and is torn and split as a result of surface tension. Part of the pellumina expands and folds into oxidation film interlayer, which is residual in alloy [12]. After such interaction, a layer of melt with low-melting point is formed in casting, which is throughout the entire casting section and forms a brittle interface after cooling. The pre-set foil in the casting mold plays a key role, which ensures the forming position of brittle surface with certainty, as it has buffers butting of two fluid flow and a great influence on the chemical mechanism. Analysis based on the chemical mechanism suggests that the increase of magnesium changes the oxidation behavior of aluminum alloy. When the content of magnesium in alloy is less than 0.005%, oxide generating in reaction is pure alumina, while the content exceeds the index, the product transforms into mixed oxide $Al_2MgO_4$, which belongs to the typical crystal structure [13], and its chemical reaction is as follows:

$$4Al + 3O_2 = 2Al_2O_3 \tag{1}$$

$$Al_2O_3 + MgO = Al_2MgO_4 \tag{2}$$

Liquid aluminum and its surface oxidation film react with AZ31 foil, forming a mixed melt composed of $Al_2MgO_4$, $Al_2O_3$ and MgO. Obvious brittle crystal boundary is produced after cooling, which directly induces the occurrence of brittle fracture.

Figure 8 shows the enlarged view of area A on fracture surface, as shown in Figure 6. The area shows the fracture characteristics at the edge of flaky fracture. The phenomenon of small part bonding exists at the top of dendritic crystal since the melting. AZ31 foil at the front of fluid flow is extruded by dendritic raft, and a kind of bonding film forms between dendritic crystals, which produces local bonding between the two overlapped alloy liquids. The atomic arrangement of this kind of casting-state organization is disorder, existing a mass of dislocations and vacancies, as well as low interatomic binding energy, which leads to perfect performance of brittle fracture and makes a contribution to the subsequent implementation of fracture splitting process.

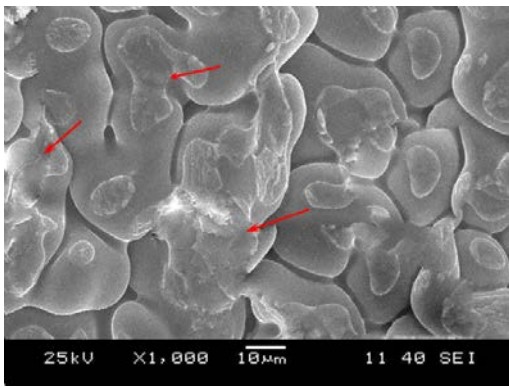

**Figure 8.** Enlarged view of area A in Figure 6.

Figure 9 shows the enlarged view of area B on the fracture surface, as shown in Figure 6. The fracture surface in this area is composed of many large aligned grains. This area has large, flat and obvious flaky fracture characteristics, which is different from ordinary characteristics of ductile fracture in basal alloy. The characteristics verify the effect of the foil. Liquid flow is extruded when overlapping on both sides of the AZ31 foil. After contacting the foil, the dendritic raft configures orderly along the vertical direction, bonding film forms on local interface where exists pellumina. When expansion load is applied, the brittle interface occurs stress concentration under vertical tension, which forces crack initiation along the interface. A micro crack passes throughout the all section expands directionally, and finally forms flaky brittle fracture.

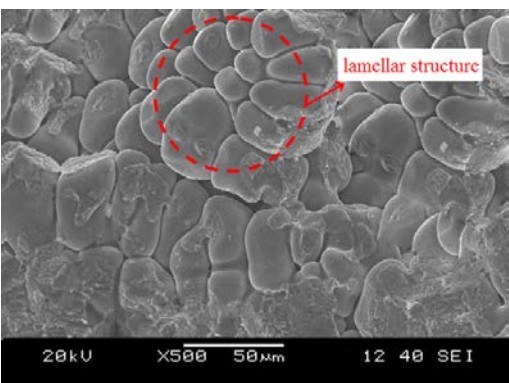

**Figure 9.** Enlarged view of area B in Figure 6.

3.2.2. Formation of Liquid Film Fracture Splitting Surface

According to fracture surface morphology shown in Figure 10, it can be seen that the fracture belongs to typical intergranular fracture. There is local bonding as well as traces of contraction liquid film between grains, and folded oxide film on grain surface. Such characteristics are beneficial to low fracture load and reduce section deformation in fracture splitting process. Further analysis suggests that the pre-set foil in casting mold oxidates severely in high-temperature aluminum liquid, which forms low-melting point mixture under common effect of pellumina at the front of metal liquid. Late in the solidification period, the solid skeleton basically forms, and linear contraction begins, mixed liquid phase is pushed to the edge of grain boundary. After grains basically complete solidification, such intergranular mixture begins to solidify. Due to lack of subsequent supplement of metal liquid, liquid film contraction occurs, as well as local bonding phenomenon between grains. In addition, the cavities stay in a stationary state after being filled with aluminum liquid, which turns the oxidation liquid film interlayer extruded into clump gradually stretch into small foils. With solidification of the alloy casting, a large number of small flaky liquid film layers form small cracks, which plays a role in cutting metal base [14].

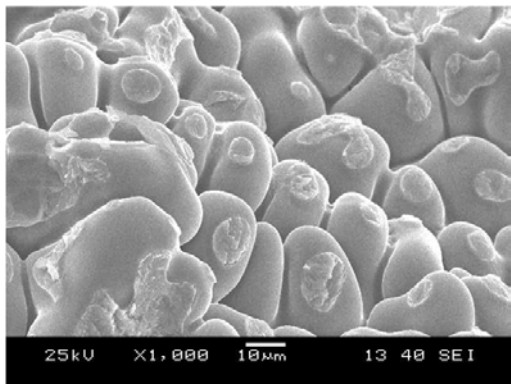

**Figure 10.** Enlarged view of area C in Figure 6.

### 3.2.3. Formation of Micro-Porosity Fracture Splitting Surface

From the microstructures shown in Figure 8 to Figure 9, it can be seen that there are micro-porosities between grains. The existence of such micro-porosities destroys the continuity of casting. With an increase in density of micro-porosities, the strength and elongation of the casting are significantly reduced. The forming of micro-porosity can be generally classified into shrinkage type and pore one. The reason for shrinkage type micro-porosity is that when metal crystallizes in casting, alloy solidifies from liquid into solid with volume shrinkage, micro-cavities are produced in crystal branches as a result of insufficient feeding of low-melting point liquid. The reason for pore type micro-porosity is that two metal liquids flowing relatively overlap in casting, which makes the outside gas stay in the gaps of crystal branches. With crystallizing, crystal branches mutually overlap to form a skeleton. Gas occurring in branches or precipitating from solidification gathers together instead of escaping. Positions occupied by the gas turn into cavities after crystallization, which finally leads to pore type micro-porosity. Micro-porosities distributing on the interface make the elastic property of alloy material discontinuous, as it has an important influence on mechanical properties of castings, and especially significantly reduces the strength and plasticity of castings in the direction which is perpendicular to the interface. When deformation becomes large, stress concentration occurs in the organization, which generally makes such micro-porosity develop into a crack source. If the micro-porosities appear in the crack extension region, several adjacent micro-porosities may turn into a small crack, and then become a large one, accelerating crack expansion and realizing casting fracture with low energy.

Scanning electron microscope is employed to analyze the composition of the casting interface. The base organization is well distributed, and the grain boundary segregation phase is thick and dense, whose major components include silicon, magnesium, which is shown in Figure 11. Analysis suggests that when setting a natural cracking surface, solute concentrates near the interface and a mass of eutectic melt with low-melting point, which is along the inter-crystalline gap and branches gap finally condenses into segregation phase. It has an important influence on the formation of the brittle interface, as silicon increases the distortion quantity of the lattice in alloy organization, as well as internal stress, while it significantly reduces the plastic deformation capacity, becoming an important factor in realizing brittle fracture.

Through analysis of chemical composition near the fracture splitting surface, it is found that the magnesium content in the interface area is obviously higher than other parts of the casting. The element distribution is shown in Figure 12. It suggests that the pre-set AZ31 foil is left near the interface layer in mold with the form of solute atoms or second phase, As a result, the occurring probability of lattice distortion, vacancy, etc. on interface increases greatly, while the binding energy of grain boundary and the stress needed for fracture crack extension are reduced, as well as the deformation degree in crack tip area. Since magnesium atoms attract electrons to its surroundings, charge density between aluminum atoms at grain boundary is reduced, leading to the decrease of binding energy and grain boundary

embrittlement. Therefore, magnesium at the grain boundary of aluminum alloy is a brittle element, which increases brittleness of grain boundary and realizes brittle fracture of aluminum alloy [15].

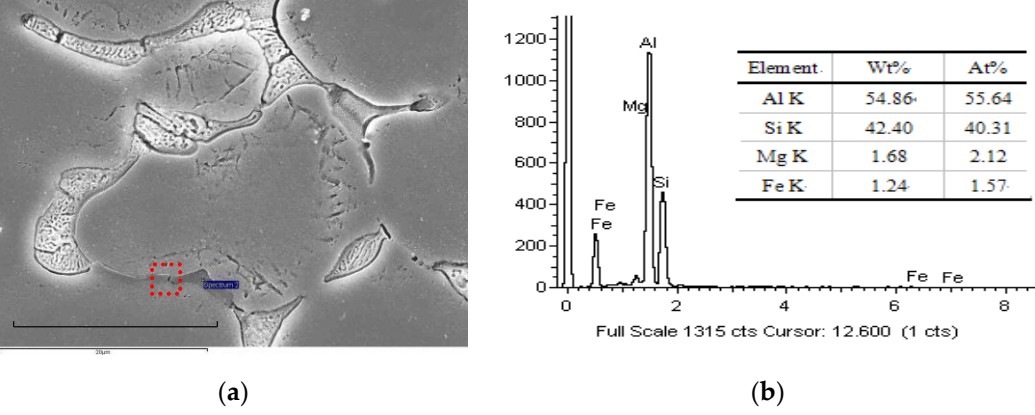

(**a**)　　　　　　　　　　　　　(**b**)

**Figure 11.** Element distribution at the grain boundary: (**a**) natural cracking surface, (**b**) elemental distribution.

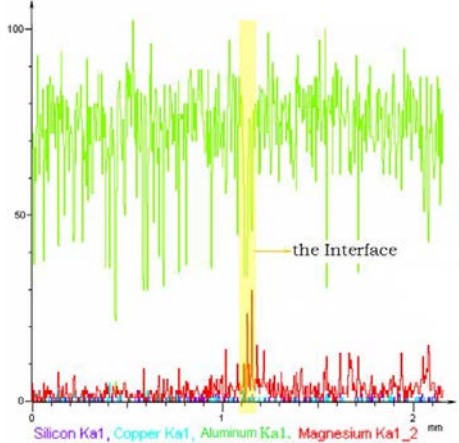

**Figure 12.** Diagram of elements distribution near the interface.

### 3.3. Comprehensive Analysis Of Natural Fracture Splitting Mechanism

Figure 13 shows the typical fracture surface morphology cracking along the fracture splitting surface, it is a comprehensive manifestation of the above fracture surface morphologies. The fracture splitting surface shows characteristics such as tiny fractures at the top of dendritic crystal, contractive liquid film between grains, and folded oxidation film on metal surface, as well as micro-porosity, which greatly increase brittle fracture performance of the material.

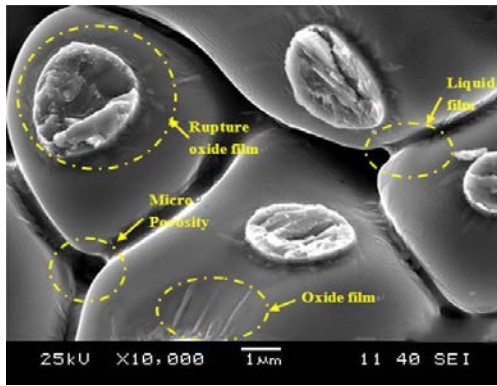

**Figure 13.** Typical fracture morphology cracking along the fracture splitting surface.

Crack position on the specimen initiates from the root of the cracking notch, and the crack expands basically along the pre-set fracture splitting surface. The fracture surface is characterized by a flaky brittle fracture, as there is residual magnesium and pellumina in the fracture surface, as well as characteristics of melt with low-melting point and micro-porosity. According to the above, casting fracture types, element composition, and microscopic characteristics of the fracture surface, as well as the typical morphology of the fracture surface cracking along the fracture splitting surface, is shown in Figure 13. Formation mechanism of the fracture splitting surface is shown in Figure 14, which can be described as follows:

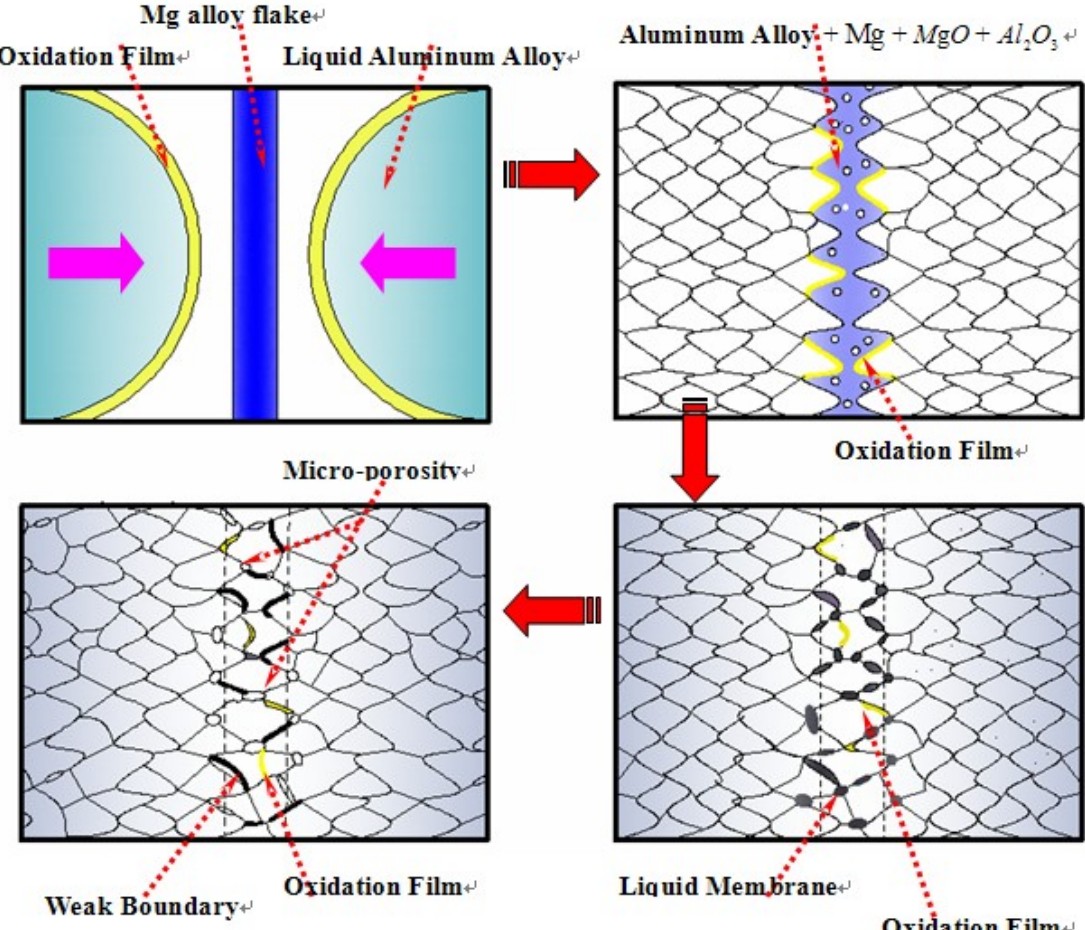

**Figure 14.** Sketch of fracture splitting surface forming process.

Two alloy liquids flowing relatively in mold overlap on each side of the AZ31 foil. An interface forms between the foil with strong oxidation in high-temperature alloy liquid and pellumina at the front of metal liquid flow, as it forms a layer of melt with low-melting point in alloy, which passes throughout the entire casting section. With growth of the crystal, the liquid phase with low-melting point is pushed to the edge of the grain boundary. Due to contraction of the surface liquid film, feeding between grains cannot go on successfully, which leads to micro-porosity. Natural fracture splitting surface finally forms under above combined effect.

Setting natural fracture splitting surface on the connecting rod is a highly creative new technique, on the premise of guaranteeing comprehensive performance of connecting rods, it enables connecting rods to achieve a brittle fracture along the interface at room temperature, the fracture surface has high strength, accurate re-meshing performance as well as locating performance [16,17].The whole process of fracture along the natural fracture splitting surface has a series of steps including evolution of micro-porosity, brittle surface failure, stress concentration, expansion and propagation of the crack and

so on [18]. Characteristics such as micro-porosity, pellumina, interface precipitation phase, etc., exist in a natural fracture splitting area on the connecting rod. The crack initiates from the root of cracking notch under expansion load, micro-cracks produce, expand and get together rapidly along the brittle surface, which finally becomes a part of the main crack. At the same time, a lot of brittle phases existing in natural fracture splitting area makes the local performance of material mismatched and deformation incompatible, making a contribution to the high brittle fracture performance of material in the interface area. Therefore, energy needed for the crack to expand stably along the fracture splitting surface is much less than the one required to expand in the basal body when the crack leaves the interface, which well realizes orientable and controllable fracture splitting, and successfully avoids phenomena such as failing to split, occurring bifurcation, and excursion on fracture surface [19,20]. The fracture surface has high strength and eliminates problems such as dropping dregs, open seam, and so on, which meets the demand of an accurate position and assembly between connecting rod body and its cover after splitting. Compared with the traditional mechanical sawing processing method, the cracking technology eliminates the mechanical sawing and grinding process of the connecting rod body/cover joint surface, but uses the natural three-dimensional concave and convex curved surface formed by the fracture to realize the three directions of the connecting rod joint surface. Precise positioning, after assembly, the connecting rod body, and the cover can be in precise contact and locked with each other. This greatly improves the bearing capacity, shear resistance, and rigidity of the connecting rod assembly. At the same time, after the second disassembly and assembly, it can still guarantee the very good engagement and assembly accuracy of the connecting rod.

### 3.4. Mechanical Performance Analysis

At the joint interface, the microhardness measurement was started on both sides. According to the obtained test data, taking the bonding interface of the two alloys as the origin of the abscissa and the microhardness value as the ordinate, the hardness distribution curve is drawn. The hardness on both sides of the interface increases first and then decreases, and the difference in hardness near the interface gradually decreases. Additionally, the excessive hardness at the interface is gradually flattening, indicating that the combination quality of the two materials is better. The hardness difference between the two sides of the as-cast structure interface does not change much. The hardness is the largest at the interface. It can be considered that there is no plastic deformation at the junction, the metal structure is dense, the performance is excellent, and the interface is fully metallurgical. The results are shown in Figure 15.

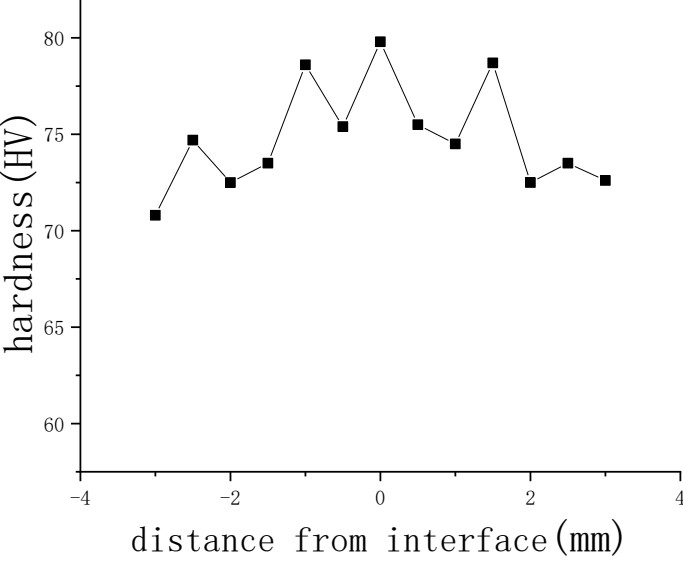

**Figure 15.** Interface hardness distribution.

## 4. Conclusions

(1) When manufacturing aluminum alloy connecting rod blanks, the natural fracture splitting surface is pre-set in the crack area on the connecting rod to make the connecting rod crack along the brittle surface in conducting fracture splitting, which realizes orientable and controllable fracture splitting at room temperature.

(2) Connecting rod blanks are manufactured by metal mold casting process. AZ31 foil with a thickness of 0.12 mm is pre-set in the mold. Under the condition that casting temperature of LD10 alloy is 720 °C, alloy liquid and the foil fully react, as a natural fracture splitting surface forms in the casting after cooling, whose quality not only ensures mechanical performance requirements of the connecting rod but also meets the demand in fracture splitting process.

(3) Fracture splitting process of casting is implemented. The specimen forms an initial crack from the root of the notch, as the crack basically expands along the pre-set fracture splitting surface, whose fracture surface is characterized by a flaky brittle fracture. Natural three-dimensional concave-convex structure left by fracture is used to guarantee the accurate reposition and assembly of the two splitting parts.

(4) Through observation of the fracture surface after splitting, it is found that there is residual magnesium and pellumina on the fracture surface, as well as characteristics of melt with low-melting point and micro-porosity. Analysis suggests that the formation of the natural facture splitting surface is caused under a combined effect of multiple mechanisms including oxidation film, liquid film shrinkage, and micro-porosity.

**Author Contributions:** Data collection, Y.Z.; data analysis, Y.Z.; writing—original draft preparation, F.Z.; writing—review and editing, F.Z.; formal analysis, F.Z; figures Y.Z.; All authors have read and agreed to the published version of the manuscript.

**Funding:** This research received no external funding.

**Acknowledgments:** Thanks for the experimental data and help provided by Jiangsu University Junkang Qian.

**Conflicts of Interest:** The authors declare no conflict of interest.

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
