# Peer review of "Setting of Natural Fracture Splitting Surface on Connecting Rod and Its Formation Mechanism"

_metals, doi:10.3390/met10050590_

Round 1

Reviewer 1 Report

The authors present a potentially interesting work. However, in my opinion, it requires quite a lot of work yet before being published. The main comments and suggestions are as follows:

  • There are no affiliations, nor correspondece details.
  • There are numerous (TOO MANY) typos all along the text.
  • English requires review.
  • Page 2, line 55: what is the "strength on fracture surface"?
  • Page 2, line 57: powder metallurgy is not a material. Please, be more specific.
  • Please, justify better the improvement of load bearing capacity (and shear capacity). Current version of the manuscript states so without ay proof or reference.
  • Figure 1: include an additional perspective (90º or from above).
  • Please, include a better explanation about the interaction between melted LD10 alloy and AZ31 foil. Now, it is too vague.
  • Quality of Figure 2 should be improved.
  • Please, provide a better definition of the geometry of the grooves.
  • Why does Figure 3 not include results for 0.24 mm?
  • Apparently, there is a contradiction between tables 3 and 4: does the foil (0.12 mm thick) experience combustion at 720ºC? Does complete combustion take place?
  • Page 6, lines 181-186: it is not clear what the authors are referring to.
  • It is recommended to include a macroscopic view of the pieces corresponding to Figures 5 and 6 of the current version.
  • Section 3.2 provides lot of reasoning, but not sound evidences and references.
  • Figure 13 must be improved.
  • The authors shoudl clearly state whether or not the introduction of the Mg foil is their idea.
  • Last, but not least: fracture splitting topic (the fracture splitting treated here) is explained in the paper through references which are 10-15 years old, and a quick look in Scientific data bases apparently indicates that this technique is not in its very best moment. Is it being abandoned by science and industry?

Author Response

1 、There are no affiliations, nor correspondece details.

This information is hidden by the editor

2 、There are numerous (TOO MANY) typos all along the text.

3 、Spelling has been improved

4、 English requires review.

Partially modified

5、 Page 2, line 55: what is the "strength on fracture surface"?

tensile strength

6、Page 2, line 57: powder metallurgy is not a material. Please, be more specific.

Powder metallurgy materials such as Cu, Cr, Zr based powder are missing during translation

7、Please, justify better the improvement of load bearing capacity (and shear capacity). Current version of the manuscript states so without ay proof or reference.

Increased the hardness test, we increased the hardness measurement value, and began to measure the microhardness at both sides of the interface. According to the obtained test data, taking the bonding interface of the two alloys as the origin of the abscissa and the microhardness value as the ordinate, the hardness distribution curve is drawn. The hardness on both sides of the interface increases first and then decreases, and the difference in hardness near the interface gradually decreases. And the excessive hardness at the interface is gradually flattening, indicating that the combination quality of the two materials is better. The hardness difference between the two sides of the as-cast structure interface does not change much. The hardness is the largest at the interface. It can be considered that there is no plastic deformation at the junction, the metal structure is dense, the performance is excellent, and the interface is fully metallurgical.

8、Figure 1: include an additional perspective (90º or from above).

The dimensions have been added, we are unable to provide a top view for the time being, but I can provide another top view of the close mold, this sample is used for our other experiments

9、Please, include a better explanation about the interaction between melted LD10 alloy and AZ31 foil. Now, it is too vague.

Two alloy liquids flowing relatively in mold overlap on each side of the AZ31 foil. An interface forms between foil with strong oxidation in high-temperature alloy liquid and pellumina at the front of metal liquid flow, there forms a layer of melt with low-melting point in alloy, which passes throughout the entire casting section. With growth of the crystal, the liquid phase with low-melting point is pushed to the edge of the grain boundary. Due to contraction of the surface liquid film, feeding between grains can not go on successfully, which leads to micro-porosity. Natural fracture splitting surface forms finally under above combined effect.

  • Quality of Figure 2 should be improved.

The quality of Figure 2 has been improved

  • Please, provide a better definition of the geometry of the grooves.

Why does Figure 3 not include results for 0.24 mm?

Added 0.24mm result

  • Apparently, there is a contradiction between tables 3 and 4: does the foil (0.12 mm thick) experience combustion at 720ºC? Does complete combustion take place?

There is an error in Table 3, the description of 0.12 mm in Table 3 has been updated,  no burning occurs at 720ºC

  • Page 6, lines 181-186: it is not clear what the authors are referring to.

It is recommended to include a macroscopic view of the pieces corresponding to Figures 5 and 6 of the current version.

The proposal includes a macro view of the parts corresponding to Figures 5 and 6 of the current version.

  • Section 3.2 provides lot of reasoning, but not sound evidences and references.

This section provides references 11, 12, 13, 14

  • Figure 13 must be improved.

The quality of Figure 13 has been improved

  • The authors shoudl clearly state whether or not the introduction of the Mg foil is their idea.

Yes, our innovative ideas. Introducing this method is our innovative idea. The origin of the idea comes from reading some literature that combines two different metals through casting. We innovatively proposed the introduction of magnesium foil into the cracking of the connecting rod. In order to solve the limitation of the cracking process on the selection of materials, improve the problems of parts tearing, cracking, slag dropping, and cross-sectional deformation in the cracking process, a new process of setting a natural cracking surface in the blank of the part Fragmentation along the preset interface during parts cracking

  • Last, but not least: fracture splitting topic (the fracture splitting treated here) is explained in the paper through references which are 10-15 years old, and a quick look in Scientific data bases apparently indicates that this technique is not in its very best moment. Is it being abandoned by science and industry?

Added Hui W, Chen S, Zhang Y, et al. Effect of vanadium on the high-cycle fatigue fracture properties of medium-carbon microalloyed steel for fracture splitting connecting rod [J]. Materials & Design, 2015, 66: 227- 234.

Kou S, Shi Z, Song W. Fracture-Splitting Processing Performance Study and Comparison of the C70S6 and 36MnVS4 Connecting Rods [J]. SAE International Journal of Engines, 2018, 11 (4): 463-474.

This technology has an important role in the field of engine connecting rod processing. Different researchers have tried to use different materials and process parameters to process connecting rod cracking.

Reviewer 2 Report

The authors present an experimental campaign for the set-up of a specific manufacturing methodology for the generation of a fracture splitting surface between a conrod and the corresponding conrod cap. A sensibility analysis is presented about some of the main manufacturing parameters. A theoretical interpretation of the different physical mechanisms involved is proposed. The approach presented is complete although some parts could be improved and investigated in more details. As a consequence, this reviewer thinks this paper is suitable to be published within this journal after the following major issues will be addressed by the authors.

Technical

  1. The author should clarify in the introduction if the proposed methodology is completely new or other applications are present of similar techniques in the pertinent literature.
  2. Details about the geometry of the specimen should be provided (dimensions, total volume…)
  3. Details about the experimental equipment should be provided (cone rod geometry, specimen constrains, application force mechanism…). Quality of Figures 2 should be improved.
  4. Results of Figure 3 and 4 (influence of Magnesium foil thickness and influence of casting temperature) should be better described. Only visual and qualitative discussion is presented. If possible, metallographic sections of the interface should be added (with eventual etching) in order to better justify/identify the best set up configuration.
  5. Comparison between base material fracture and fracture surface with AZ31 foil (section 3.1) should be described in more details. In particular, add quantitative information about the different external load necessary to promote the split mechanism in the two configurations.
  6. If possible, XRD analysis should be performed of the split surface to confirm the presence of Al2MgO4 (section 3.2 (1)).
  7. The surface morphology shown in Figures 5, 7, 8 and 9 seems to present limited contact surface with respect to the counter surface. In other terms, the process seems to create what in metal casting is called a "cold joint", as if the two surfaces were separated in most areas before the rupture. As a consequence, the match between peaks and valleys of the two surfaces needs to be proven. The latter is needed for the success of the technique. Please add a discussion about this point since the perfect complementarity between the meting surfaces is necessary for the correct operation of the coupling (i.e. peaks of one surface have to correspond to valleys of the counterpart. Images seem to suggest a sort of peak to peak contact).
  8. Hardness measurements of the split surface should be provided (in comparison with base material), in order to prove the interface is capable to sustain the frictional behaviour of the achieved structure and to exclude that the displayed structure undergo plastic deformation during tightening and operation.

Author Response

1、The author should clarify in the introduction if the proposed methodology is completely new or other applications are present of similar techniques in the pertinent literature.

The author should clarify in the introduction whether the proposed method is completely new or there are other applications of similar techniques in the relevant literature.

A similar application citation paper has been added [18-19], the technology is applied to the processing of automobile engine connecting rods, this process is our innovative idea, In order to solve the limitation of the cracking process on the selection of materials, improve the problems of parts tearing, cracking, slag dropping, and cross-sectional deformation in the cracking process, a new process of setting a natural cracking surface in the blank of the part Fragmentation along the preset interface during parts cracking

2、Details about the geometry of the specimen should be provided (dimensions, total volume…)

Details about the experimental equipment should be provided (cone rod geometry, specimen constrains, application force mechanism…). Quality of Figures 2 should be improved.

The size has been marked on Figure 2, the sample is 8 cm long, 3 cm wide, and 2 cm high. The quality of Figure 2 has been improved. The mallet is cylindrical and uses a gap fit.

3、Results of Figure 3 and 4 (influence of Magnesium foil thickness and influence of casting temperature) should be better described. Only visual and qualitative discussion is presented. If possible, metallographic sections of the interface should be added (with eventual etching) in order to better justify/identify the best set up configuration.

We have tested that if the temperature is heated at 0.1mm, it is easy to cause rapid combustion, making the entire process uncontrollable. If the thickness of the aluminum foil is too large and more than 0.2mm, the edges will not be fused, so a thickness between 0.1-0.2mm is a reasonable thickness. 0.12 is selected in this article mm thickness.

4、Comparison between base material fracture and fracture surface with AZ31 foil (section 3.1) should be described in more details. In particular, add quantitative information about the different external load necessary to promote the split mechanism in the two configurations.

If possible, XRD analysis should be performed of the split surface to confirm the presence of Al2MgO4 (section 3.2 (1)).

Two macro pictures have been added. We have added two fracture photos. Figure 5 is the fracture characteristics of the casting with natural cracking surface after cracking. The fracture morphology is flaky brittle fracture, and there is plastic deformation in the edge of the flaky fracture. The area occupied by the area is relatively small, showing obvious brittle fracture characteristics. The micro-fracture after cracking of the base metal casting as a control is shown in Fig. . The fracture has obvious plastic deformation, and there are many ductile tear marks, which are typical ductile fractures. The quantitative information of different external loads required to promote the split mechanism is not the key to the discussion in this article. This article mainly discusses the fracture features and morphological features

5、The surface morphology shown in Figures 5, 7, 8 and 9 seems to present limited contact surface with respect to the counter surface. In other terms, the process seems to create what in metal casting is called a "cold joint", as if the two surfaces were separated in most areas before the rupture. As a consequence, the match between peaks and valleys of the two surfaces needs to be proven. The latter is needed for the success of the technique. Please add a discussion about this point since the perfect complementarity between the meting surfaces is necessary for the correct operation of the coupling (i.e. peaks of one surface have to correspond to valleys of the counterpart. Images seem to suggest a sort of peak to peak contact).

The phenomenon you mentioned can be eliminated by heat treatment. We have done similar experiments before. As shown in Figure 6 (b), we find that the quasi-cleaved surfaces are arranged neatly and belong to obvious brittle fracture. After the forging process, the Si element in A390 changes from brittle phases such as large primary crystal silicon and short rod-shaped eutectic silicon to fine and uniform silicon particles, which are relatively evenly distributed in the aluminum matrix. The silicon particles together form a quasi-cleaved surface, so the fracture surface is smooth and flat, and there are basically no cracks, which avoids the generation of cracks and particles during the cracking of the cast slab, which greatly improves the quality of the cracked surface.

6、Hardness measurements of the split surface should be provided (in comparison with base material), in order to prove the interface is capable to sustain the frictional behaviour of the achieved structure and to exclude that the displayed structure undergo plastic deformation during tightening and operation.

We increased the hardness measurement value, and began to measure the microhardness at both sides of the joint interface. According to the obtained test data, taking the bonding interface of the two alloys as the origin of the abscissa and the microhardness value as the ordinate, the hardness distribution curve is drawn. The hardness on both sides of the interface increases first and then decreases, and the difference in hardness near the interface gradually decreases. And the excessive hardness at the interface is gradually flattening, indicating that the combination quality of the two materials is better. The hardness difference between the two sides of the as-cast structure interface does not change much. The hardness is the largest at the interface. It can be considered that there is no plastic deformation at the junction, the metal structure is dense, the performance is excellent, and the interface is fully metallurgical.

Reviewer 3 Report

Dear Authors,

  1. (Section 2.3) The suitable AZ31 foil thickness may be affected by either the mold cavity size or the casting temperature. The thickness of 0.12 mm may a suitable one only for a narrow range of the casting conditions. Please add the experimental results or discussions about the effects of the related casting conditions.
  2. (Figure 4 & Table 4) In Figure 4, the temperatures are not in ascending order and it may cause difficulties to readers. It is strange that as the casting temperature changes from 670 deg to 720 deg, the fusion trace becomes smaller. Please explain the phenomena or show the related photos.
  3. Please use the same degree of magnification for photos to be compared to each other. For example, Figure 5 & 6, Figure 7, 8 & 9. In Figure 10, no degree of maginication is shown.
  4. From Figure 5 to 12, photos for the suitable thickness and casting temperature are shown. If possible, add photos for not suitable conditions for comparison. 

Author Response

1、(Section 2.3) The suitable AZ31 foil thickness may be affected by either the mold cavity size or the casting temperature. The thickness of 0.12 mm may a suitable one only for a narrow range of the casting conditions. Please add the experimental results or discussions about the effects of the related casting conditions.

We have tested that if the temperature is heated at 0.1mm, it is easy to cause rapid combustion, making the entire process uncontrollable. If the thickness of the aluminum foil is too large and more than 0.2mm, the edges will not be fused, so a thickness between 0.1-0.2mm is a reasonable thickness. 0.12 is selected in this article mm thickness.

2、(Figure 4 & Table 4) In Figure 4, the temperatures are not in ascending order and it may cause difficulties to readers. It is strange that as the casting temperature changes from 670 deg to 720 deg, the fusion trace becomes smaller. Please explain the phenomena or show the related photos.

According to the relevant composite theory, the entire process of bimetallic interface formation includes three stages: physical contact formation, contact surface activation, and element diffusion. High temperature can melt the metal and promote the full contact of the two alloys. At high temperature, infiltration and diffusion occur, so that atoms between different elements Migration occurs to form a strong metallurgical bond. [13]

3、Please use the same degree of magnification for photos to be compared to each other. For example, Figure 5 & 6, Figure 7, 8 & 9. In Figure 10, no degree of maginication is shown.

From Figure 5 to 12, photos for the suitable thickness and casting temperature are shown. If possible, add photos for not suitable conditions for comparison.

We have adjusted the order of images according to the convenience of readers. Figures 5 and 6 have used the same scale, and the magnification has been added to Figure 10. Figures 7 and 8 、 9 use different magnification photos for better interpretation.

We added two fracture photos. Figure 5 is the fracture characteristics of the casting with natural cracking surface after cracking. Its fracture morphology is flaky brittle fracture. The edge of the flaky fracture is locally plastically deformed, but the area occupied is small. , Showing obvious brittle fracture characteristics. The micro-fracture after cracking of the base metal casting as a control is shown in Fig. 6. The fracture has obvious plastic deformation, and there are many ductile tear marks, which are typical ductile fractures.

(A) LD10 base material macro fracture (b) foil added casting macro fracture

                       Figure 5 Macro fracture after casting cracking

Round 2

Reviewer 1 Report

The authors have vaguely answered or commented the different aspects of the first review. Mayor concerns remain the new version (English, novelty, etc). Moreover, they argue that the increase in the load bearing capacity is justified by the increase on hardness. This is not necessary true: an increase in hardness probably implies an increase in tensile strength, but it also reduces ductility and fracture toughness. In the presence of defects, the overall load bearing capacity can, therefore, be lower.

My recommendation is rejection, encouraging authors to rearrange and improve the paper, and try again with another journal.

Author Response

Hardness is a key parameter of a material's ability to resist deformation due to local pressure, and there is a direct correspondence between the hardness and strength of alloy materials. The hardness of the material directly reflects the continuous changes in the structure and properties of the bimetallic interface of the bimetal forging billet.

Reviewer 2 Report

The authors addressed most of the points raised by the present reviewer. Nevertheless, the following minor issues remain before the paper is suitable for publication.

Technical

  1. Quality of Figure 3b should be improved. Caption of Figure 3 has to be adjusted ((c) is missing and 0.12 is missing). At the same time, description added at the beginning of Section 2.3 sounds quite confusing: “We have tested that if the temperature is heated at 720°C and an AZ31 foil with a thickness of 0.1mm is considered, it is easy to cause rapid combustion, making the entire process uncontrollable. If the thickness of the aluminum (AZ31?) foil is too large and more than 0.2mm, the edges will not be fused, so a thickness between 0.1 0.2mm is a reasonable thickness. A foil thickness of 0.12mm is selected in this article.
  2. Figure 6 seems not updated in the manuscript (Figure 6(a) and 6(b) are missing).
  3. Caption of Figure 15 is missing. Please add a schematic in order to clarify the meaning of “Distance” in the x-axis.
  4. In the introduction Authors state “Compared to traditional saw cutting method, fracture splitting technique cancels mechanical sawing and grinding process of joint surface on connecting rod body or its cover, it utilizes natural three-dimensional concave-convex surface forming through fracture to obtain accurate position of the joint surface on connecting rod in three directions.” Is this concave-convex surface guaranteed also by the proposed method or mechanisms summarized in Figure 14 alter the formation of this natural and complementary concave-convex surface? Please add a comment in the manuscript. The following discussion has been added by authors in the covering letter “The phenomenon you mentioned can be eliminated by heat treatment. We have done similar experiments before. As shown in Figure 6 (b), we find that the quasi cleaved surfaces are arranged neatly and belong to obvious brittle fracture. After the forging process, the Si element in A390 changes from brittle phases such as large primary crystal silicon and short rod shaped eutectic silicon to fine and uniform silicon particles, which are relatively evenly distributed in the aluminum matrix. The silicon particles together form a quasi cleaved surface, so the fracture surface is smooth and flat, and there are basically no cracks, which avoids the generation of cracks and particles during the cracking of the cast slab, which greatly improves the quality of the cracked surface.” What’s the heat treatment process the authors refer to? Is the process described casting or forging?

Author Response

Technical

1、Quality of Figure 3b should be improved. Caption of Figure 3 has to be adjusted ((c) is missing and 0.12 is missing). At the same time, description added at the beginning of Section 2.3 sounds quite confusing: “We have tested that if the temperature is heated at 720°C and an AZ31 foil with a thickness of 0.1mm is considered, it is easy to cause rapid combustion, making the entire process uncontrollable. If the thickness of the aluminum (AZ31?) foil is too large and more than 0.2mm, the edges will not be fused, so a thickness between 0.1 0.2mm is a reasonable thickness. A foil thickness of 0.12mm is selected in this article.”

Improved the quality of Figure 3b and added the title C of Figure 3,The chaotic expression is deleted, and only the comparison of the three thicknesses is kept. Since the focus of this study is not on the choice of thickness, there is no discussion of thermophysical and chemical properties. Such experiments are complicated

2、Figure 6 seems not updated in the manuscript (Figure 6(a) and 6(b) are missing).

Figure 6 is not updated, and the description is deleted by mistake. Figures 6 (a) and 6 (b) are actually deleted. In fact, Figures 5 (a) and (b) are added

3、Caption of Figure 15 is missing. Please add a schematic in order to clarify the meaning of “Distance” in the x-axis.

Increased the meaning of X-axis distance, distance from interface

4 In the introduction Authors state “Compared to traditional saw cutting method, fracture splitting technique cancels mechanical sawing and grinding process of joint surface on connecting rod body or its cover, it utilizes natural three-dimensional concave-convex surface forming through fracture to obtain accurate position of the joint surface on connecting rod in three directions.” Is this concave-convex surface guaranteed also by the proposed method or mechanisms summarized in Figure 14 alter the formation of this natural and complementary concave-convex surface? Please add a comment in the manuscript.

The expression has been added: Compared with the traditional mechanical sawing processing method, the cracking technology eliminates the mechanical sawing and grinding process of the connecting rod body / cover joint surface, but uses the natural three-dimensional concave and convex curved surface formed by the fracture to realize the connecting rod joint surface Accurate positioning in three directions, after assembly, the connecting rod body and the cover can be in precise contact and locked with each other, which greatly improves the bearing capacity, shear resistance, and rigidity of the connecting rod assembly. At the same time, after the second disassembly and assembly, it can still guarantee the very good engagement and assembly accuracy of the connecting rod.

The following discussion has been added by authors in the covering letter “The phenomenon you mentioned can be eliminated by heat treatment. We have done similar experiments before. As shown in Figure 6 (b), we find that the quasi cleaved surfaces are arranged neatly and belong to obvious brittle fracture. After the forging process, the Si element in A390 changes from brittle phases such as large primary crystal silicon and short rod shaped eutectic silicon to fine and uniform silicon particles, which are relatively evenly distributed in the aluminum matrix. The silicon particles together form a quasi cleaved surface, so the fracture surface is smooth and flat, and there are basically no cracks, which avoids the generation of cracks and particles during the cracking of the cast slab, which greatly improves the quality of the cracked surface.” What’s the heat treatment process the authors refer to? Is the process described casting or forging?

Figure1 SEM image of cracked area after simulated cracking of aluminum alloy bimetallic connecting rod

The original figure 1 is an enlarged view of the A390 fracture in the cracking zone of the slab. Judging from the morphology of the fracture, there are many bright and flat quasi-cleavage surfaces on the fracture surface. The transitional steps between the facets form many sharp edges and corners. The angular tip easily falls off to form tiny particles during the fracture process. If these particles are not removed in time during the cracking process, the inclusion of these particles between the cracking surfaces will aggravate the wear of the cross section and reduce its secondary assembly performance.

After forging

As shown in Figure 2, it is found that the quasi-cleaved surfaces are arranged neatly and belong to obvious brittle fracture. After the forging process, the Si element in A390 changes from brittle phases such as large primary crystal silicon and short rod-shaped eutectic silicon to fine and uniform silicon particles, which are relatively evenly distributed in the aluminum matrix. The silicon particles together form a quasi-cleaved surface, so the fracture surface is smooth and flat, and there are basically no cracks, which avoids the generation of cracks and particles during the cracking of the cast slab, which greatly improves the quality of the cracked surface.

Figure 3 after heat treatment

As shown in Fig. 3, the solid solution treatment makes the bimetallic slab dense in structure and excellent in performance, and the interface metallurgy is fully combined. In this experiment, the effect of different temperatures and time on the structure and mechanical properties of bimetallic castings was studied. The optimal temperature was 500 ℃ and the time was 4h. The heat treatment can improve the structure and mechanical properties of bimetallic composites. The production of bimetallic composite cracking connecting rod is of guiding significance.

Reviewer 3 Report

Dear Authors,

Thank you for responding to review comments well.

Author Response

Thanks for your valuable comments and suggestions